# Cancer Stem Cell Markers in Rhabdomyosarcoma in Children

**DOI:** 10.3390/diagnostics12081895

**Published:** 2022-08-04

**Authors:** Joanna Radzikowska, Anna M. Czarnecka, Teresa Klepacka, Magdalena Rychłowska-Pruszyńska, Anna Raciborska, Bożenna Dembowska-Bagińska, Maciej Pronicki, Andrzej Kukwa, Wojciech Fendler, Urszula Smyczyńska, Wojciech Kukwa, Antoni Krzeski

**Affiliations:** 1Department of Otorhinolaryngology, Faculty of Medicine and Dentistry, Medical University of Warsaw, Stepinska 19/25, 00-739 Warsaw, Poland; 2Department of Soft Tissue/Bone Sarcoma and Melanoma, Maria Sklodowska-Curie National Research Institute of Oncology, Roentgena 5, 02-781 Warsaw, Poland; 3Department of Experimental Pharmacology, Mossakowski Medical Research Centre, Polish Academy of Sciences, Pawinskiego 5, 02-106 Warsaw, Poland; 4Department of Pathology, Institute of Mother and Child, Kasprzaka 17a, 01-211 Warsaw, Poland; 5Department of Oncology and Surgical Oncology for Children and Youth, Institute of Mother and Child, Kasprzaka 17a, 01-211 Warsaw, Poland; 6Department of Pediatric Oncology, The Children’s Memorial Health Institute, Dzieci Polskich 20, 04-730 Warsaw, Poland; 7Department of Pathology, The Children’s Memorial Health Institute, Dzieci Polskich 20, 04-730 Warsaw, Poland; 8Department of Otolaryngology and Head and Neck Diseases, School of Medicine, University of Warmia and Mazury, Warszawska 30, 10-082 Olsztyn, Poland; 9Department of Biostatistics and Translational Medicine, Medical University of Lodz, Mazowiecka 15, 92-215 Lodz, Poland

**Keywords:** rhabdomyosarcoma, cancer stem cells, sarcoma stem cells

## Abstract

(1) Background: The aim of the present study was to assess the cancer stem cell (CSC) markers CD24, CD44, CD133, and ALDH1A1 in rhabdomyosarcoma (RMS) in children and to define their prognostic role in this group of patients. (2) Methods: The study material was archival tissue specimens collected from 49 patients under 18 years of age and who had been diagnosed with RMS. Immunohistochemistry (IHC) was used to evaluate the expression of the selected CSC markers in the tumor tissue. Expression was evaluated using a semiquantitative IRS scale based on the one developed by Remmele and Stenger and was correlated with the clinical and pathomorphological parameters of prognostic importance in RMS. (3) Results: Expression of the selected CSC markers CD24, CD44, CD133, and ALDH1A1 was demonstrated in 83.7%, 55.1%, 81.6%, and 100% of the RMS patients, respectively. The expression of all of the assessed CSC markers was statistically significantly higher in the study group versus the control group. No significant correlation was found between the expression of the selected CSC markers and clinical and pathological prognostic factors that were analyzed. The expression of the CSC markers did not have a significant influence on RMS survival rates. (4) Conclusions: The results of the conducted study confirm the expression of selected CSC markers in rhabdomyosarcoma tissue in children. The study did not support the prognostic relevance of the expression of any of the assessed CSC markers. However, further studies are needed to fully understand the relevance of the selected CSC markers in RMS carcinogenesis.

## 1. Introduction

Rhabdomyosarcoma (RMS) is the most common type of soft tissue sarcoma (STS) in children and the third most prevalent childhood extracranial solid tumor after neuroblastoma and Wilms tumors [1]. The incidence of RMS is estimated at 4.5% of all cases of childhood cancer [1]. More than 50% of cases are diagnosed in the first decade of life [2,3]. While four distinct subtypes of rhabdomyosarcoma can be distinguished: embryonal (ERMS), alveolar (ARMS), pleomorphic, and sclerosing/spindle cell [4], in children, two main subtypes of RMS: ERMS (60% of all RMS cases) and ARMS (20% of all RMS cases), are diagnosed the most often [5]. The subtype-dependent 5-year survival varies from 35% to 90% [6]. The carcinogenesis of RMS has rarely been discussed in the literature, as the rarity of RMS cases makes it difficult to enroll a sufficiently large study group. Histopathological diagnostics, which are based on IHC as well as cytogenetic tests, require experienced pathologists and the results to be verified in reference centers.

Over the last few decades, the prognosis for children with localized RMS has significantly improved, with a 5-year overall survival rate of >70% [1]. At the same time, despite aggressive combination therapy, no further significant improvements have been made for the treatment of children with high-risk disease or recurrent disease (5-year survival <30% and 17%, respectively) [2]. Favorable prognostic factors include a primary tumor site in the orbit and tumors that are located in the non-parameningeal head and neck region and genitourinary sites, with the exception of the bladder and prostate; from patients aged 1–9 years; a lack of distant metastases at diagnosis; a clinical stage based on the classification of the primary procedure; a maximum tumor diameter ≤5 cm; and embryonal histology [1,7]. According to a report from the COG (The Children’s Oncology Group), PAX-FOXO1 fusion results in unfavorable outcomes in children with RMS [8]. In contrast to ARMS, fusion positivity is extremely rare in ERMS. The genetic abnormalities that are observed in ERMS are more diverse. While the presence of PAX-FOXO1 fusion genes correlates with a worse prognosis and while fusion-negative ARMS has a similar outcome to ERMS, molecular evaluation has become more significant in predicting outcomes, and new molecular markers are needed [4].

Theories related to cancer stem cells and sarcoma stem cells (SSC) have gained more and more interest from the medical community over the last dozen years. The CSC hypothesis was born on the basis of research concerning the molecular processes underlying the neoplastic cell resistance to conventional oncological treatment and relapse despite initial remission. Cancer stem cells, also known as “tumor-initiating cells” (TIC) are generally defined as a small subpopulation of tumor cells with stem cell-like properties that are related to tumor initiation, therapeutic resistance, disease relapse, and metastasis. The presence of CSC within the tumor is of significant clinical importance, as they constitute a reservoir of cells that are resistant to conventional oncological treatment and that are responsible for tumor progression, relapse, and metastasis [9]. Although there is still no single universal CSC marker that has been found, several methods allow the identification and isolation of subpopulations of cells whose oncogenic potential is subsequently confirmed by in vitro and in vivo tests. The detection of surface markers, also called CD molecules (cluster of differentiation), that act as receptors or ligands in signaling cascades or that participate in other cell processes such as adhesion and migration is one of the commonly used methods. CD133, CD44, and CD24 are considered to be the best-known CSC markers in solid tumors [10], and together with the family of cytoprotective enzymes, ALDH, have been widely analyzed and are well-known CSC markers of potential clinical importance. This study aimed to evaluate the expression of the selected stem cell markers CD24, CD44, CD133, and ALDH1A1 in rhabdomyosarcoma in children and to determine their prognostic significance in this disease.

## 2. Materials and Methods

### 2.1. Study Population

Forty-nine RMS patients who were under 18 years of age at diagnosis and who started treatment between 1/2000 and 12/2016 at the Department of Pediatric Oncology at the Children’s Memorial Health Institute in Warsaw (Poland) and the Department of Oncology and Surgical Oncology for Children and Youth at the Institute of Mother and Child in Warsaw (Poland) were included in the study. The last follow-up was on 2/15/2021. Primary tumor samples were obtained by biopsy or surgery in treatment-naïve cases prior to chemotherapy or radiation therapy. Normal striated muscle tissue to be used as a control was obtained from 18 sarcoma-free individuals under 18 years of age following tonsillectomy due to sleep-disordered breathing or after thyroid-lingual cyst resection. The study was approved by the local bioethics committee of the Medical University of Warsaw (AKME/64/13).

### 2.2. Analyzed Clinical Parameters

Age at time of diagnosis, sex, histopathological subtype (ERMS vs. ARMS), primary tumor site (favorable vs. unfavorable), tumor size (a (≤5 cm) vs. b (>5 cm)), T traits, regional lymph nodes involvement, the presence of distant metastases, and disease stage according to the pretreatment TNM staging for childhood RMS as defined by the Intergroup Rhabdomyosarcoma Study Group [11] were analyzed. Orbit, the head and neck excluding the parameningeal region, and the genitourinary tract excluding the bladder and prostate were considered prognostically favorable tumor sites. For the overall survival (OS) analysis, the time from diagnosis to death from any cause or until the last follow-up was calculated.

### 2.3. Immunohistochemistry

For immunohistochemical staining, slides that were 3 µm thick were stained with EnVision FLEX Hematoxylin (DAKO, K8008). The Dako PT Link Pre-Treatment Module was used for dewaxing, hydration, and heat-induced epitope retrieval. Anti-CD24 (Bioss, bs-0528R, diluted 1:200), anti-CD44 (DAKO, M7082, diluted 1:50), anti-CD133 (Biobryt, orb18124, diluted 1:200), and anti-ALDH1A1 (Santa Cruz Biotechnology, sc-374076, diluted 1:500) primary antibodies were used. The EnVisionTM FLEX + detection system with horseradish peroxidase (DAKO, K8002) was used. IHC staining was assessed by two independent pathologists. The semiquantitative IRS (immunoreactive score) scale based on one developed by Remmele and Stegner was used to assess the expression of the CSC markers [12].

### 2.4. Statistical Analysis

Before the cellular markers were analyzed, the baseline descriptive statistics of the patient subgroups (the control and cancer groups, cancers of distinct stages, etc.) were calculated. Continuous characteristics (age, marker expressions) were compared between groups using the Mann–Whitney U test, while categorical ones (gender, cancer stages, histologic type distribution) were compared using Fisher’s exact test. The staining intensity and the expression of the cellular markers were compared between tissues from diverse groups using the Mann–Whitney U test or generalized linear models (GLMs). GLMs were used whenever the analyzed subgroups differed significantly according to important covariates (age, frequency of histologic subtypes of RMS) to correct for their effects. The frequencies of the IRS scores were compared using the two-tailed Fisher’s exact test or using logistic regression, with the latter being applied again as a means to correct for the effects of the covariates. Survival was analyzed using the Kaplan–Meier method with the log-rank test for categorical predictors and with Cox regression for continuous ones. For visualization, some of the continuous variables were dichotomized at the median so that the Kaplan–Meier curves could be presented.

## 3. Results

### 3.1. Study Population

The study group included 19 females and 30 males (Table 1). The median age at diagnosis in the RMS group was 4.8 years (interquartile range (IQR) 2.3–8.7 years). ERMS was diagnosed in 30 cases, while ARMS was diagnosed in 19 cases. In 17 patients, the primary tumor site was diagnosed as being in a favorable localization, with an unfavorable tumor site being diagnosed in the remaining 32 patients. In 67% of all cases, the primary tumor size exceeded 5 cm in diameter at diagnosis. In total, 15 patients (31%) were diagnosed with stage 1 disease, 2 (4%) were diagnosed with stage 2 disease, 15 were diagnosed with stage 3 disease, and 17 (34%) were diagnosed with stage 4 disease. The median follow-up was 6 years and 1 month (6 months to 22 years and 10 months). At the time of analysis, 22 patients died due to RMS disease progression.

### 3.2. CD24, CD44, CD133, and ALDH1A1 Expression in RMS

The IRS scores for all of the assessed CSC markers (CD24, CD44, CD133, and ALDH1A1) were statistically significantly higher in the RMS tumors than they were in the normal tissues from the control group. The percentage of positively stained cells as well as the staining intensity for all of the selected CSC markers was significantly higher in the RMS tumors than it was in normal muscle (Figure 1). For CD24 and CD133, expression was observed in the cell membrane and cytoplasm, while CD44 was only expressed in the cell membrane, and ALDH1A1 was only expressed in the cytoplasm.

### 3.3. Expression of Stem Cell Markers in Different Disease Stages

For the CSC markers that were analyzed, no statistically significant differences were found in the IRS scores, marker expression, or intensity of the IHC staining between the T1 and T2 tumors (Table 2). Moreover, the IRS scores for CD24, CD44, CD133, and ALDH1A1 and the expression of the CSC markers, and the intensity of the IHC staining did not correlate with the disease stage (Table 3).

### 3.4. Overall Survival Prognostic Factors

The overall survival of the patients with stage 4 disease (*p* = 0.0045), a tumor size >5 cm (*p* = 0.0134), N1 stage (*p* = 0.0168), distant metastases (*p* = 0.0006), and alveolar histology (*p* = 0.0279) (Table 4) (Appendix A) was significantly shorter. Age was also proven to be a statistically significant factor influencing prognosis (*p* = 0.0095; HR = 1.11) (Table 5). In the group of children with stage 4 disease, ARMS was diagnosed more frequently (*p* = 0.0127) and these patients were significantly older (*p* = 0.0169). (Appendix A).

The expression of CD24, CD44, CD133, and ALDH1A1 did not significantly correlate with OS, neither by the percentage of positively stained cells nor by the intensity of the expression on/in the cell (Table 4 and Table 5) (Figure 2, Appendix A).

## 4. Discussion

According to the presented study, statistically significant higher expression of all of the assessed CSC markers was found in the RMS compared to in the normal striated muscle tissue. CD133 was one of the first CSC markers to be analyzed in sarcoma patients. Sana et al. were the first to demonstrate the expression of CD133 in RMS tissue (in biopsy material from seven patients, including one with recurrent disease) and five RMS cell lines. Using the IHC method, they observed a small subpopulation of neoplastic cells with reaction intensities varying from weak to strong [13]. Walter et al. identified the CSC population in ERMS cell lines. For CD133+ rhabdospheres they confirmed increased oncogenic potential in functional tests and increased resistance to conventional chemotherapeutic agents. CD133 expression in neoplastic tissue was found in tissue material from 76 patients diagnosed with ERMS who were enrolled in the CWS95 study. Immunofluorescence staining for CD133 revealed a positive reaction in 80% of the assessed RMS cells [14]. Pressey et al. confirmed the expression of CD133 in 12.7% to 53.5% of RMS cells. The subpopulation of low-differentiated RMS CD133+ cells was capable of spheroids formation and was resistant to conventional chemotherapy. [15]. Zambo et al. assessed CD133 as well as nestin and ABCG2 in expression in pediatric sarcomas, confirming the presence of CD133+ cells in 14 of 24 RMS cases [16]. CD133, the first member of the prominin family (prominin-1), is a pentaspan cholesterol-binding membrane glycoprotein with a total molecular weight of 120 kDa [17,18,19]. CD133 is selectively exposed in plasma membrane protrusions such as microvilli and cilia [20]. The biological function of CD133 is not yet fully understood. The concentration of CD133 in a region where there are cytoplasmic protrusions suggests that it plays a role in the formation and regulation of the cell membrane topology [21]. Interactions with plasma membrane lipids indicate the structural role of CD133 and suggest its participation in signal transduction [22]. According to Marzesco et al., CD133 expression may be important for maintaining the ability of stem cells to differentiate, while the release of glycoprotein initiates the differentiation process of neuroepithelial cells [23].

Humphrey et al. analyzed the expression of the CD44s glycoprotein in 28 RMS cases and found positive expression, with at least 60% of the tumor cells being positively stained. They found that ARMS mostly does not express CD44s, contrary to the majority of ERMS cases [24]. Similarly, Saxon et al. observed no CD44 expression in ARMS; however, positive expression was found in 4 out of 12 cases [25]. Heerema-McKenney et al. also noticed no CD44 expression in the majority of ARMS cases. The assessed parameter was the number of stained cells, with a cut-off point of 5% [26]. The transmembrane glycoprotein CD44, cell adhesion molecule (CAM), binds to several ligands, including the most specific one, hyaluronic acid (HA), and other extracellular matrix ligands (ECM) such as osteopontin, integrin, fibronectin, laminin, matrix metalloproteinases (MMPs), and collagen [27,28]. CD44 mediates cell−extracellular matrix and cell−cell interactions, thereby maintaining the integrity of organs and tissues. The HA−CD44 complex leads to cell signaling that enhances the adhesion, aggregation, proliferation, and migration of many cell types (including lymphocytes, macrophages, and fibroblasts). CD44 is involved in both physiological processes (embryogenesis, angiogenesis, inflammation, wound healing, and apoptosis) as well as carcinogenesis [27,29]. HA−CD44 interactions, which are mediated by anacrin, RhoA (Ras homolog gene family member A), Rac1 (Ras-related C3 botulinum toxin substrate 1), and CDC42 (cell division cycle 42) as well as receptor tyrosine kinases (RTKs) activate signaling pathways leading to structural changes in the cytoskeleton, ECM degradation, and the release of cytokines that facilitate adhesion, migration, invasion and the growth of neoplastic cells [30]. The alternative-splicing process of the gene encoding the CD44 receptor and numerous post-translational modifications results in the formation of different isoforms, called variants (CD44v1-CD44v10), which are different in terms of both structure and function, that make up the CD44 protein family [29].

Little is known about CD24 expression in sarcomas. There is no literature on CD24 expression in RMS available according to the authors’ knowledge. The presented study is probably the first report on the expression of this glycoprotein in rhabdomyosarcoma. CD24 is a mucin-type glycosylphosphatidylinositol-linked cell surface protein with a molecular weight of 25–75 kDa [31]. It mediates cell−cell and cell−ECM interactions [31,32]. By binding to P-selectin, a cellular adhesion molecule that is present on the surface of activated vascular endothelial cells and platelets, CD24 participates in cell adhesion and migration [33]. As a P-selectin ligand CD24 plays also a significant role in the oncogenesis of many types of cancers, allowing neoplastic cells to roll on the endothelial cells and metastasize [34]. CD24 also affects the proliferation of cancer cells and their adhesion to fibronectin, collagen, and laminin [35].

The growing amount of literature considering increased ALDH activity in many cancer types confirms the essential cytoprotective role of this enzyme for tumor cell survival and disease progression [36]. There is a lot of proof that different ALDH isoforms are responsible for the increased metastatic potential in different types of cancer [37]. The increased expression and activity of the ALDH1 isoform are associated with a worse prognosis in some neoplasms, including sarcomas [37,38,39]. Lohberger et al. found high ALDH1 activity in a small percentage of cells with stem-like properties derived from five sarcoma cell lines [40]. Martinez-Cruzado et al. observed a gradual increase in the expression and activity of ALDH1 (especially ALDH1A1 and ALDH1A3) in the CSC subpopulation that showed increased oncogenic potential during tumor progression [39]. This relationship indicates the utility of ALDH1 as a CSC marker in sarcomas and also suggests its potential prognostic role [39]. Little is known about the utility of ALDH1 as a CSC marker in pediatric oncology. Nakahata et al. selected neoplastic cells with high ALDH1 activity and stem-like properties in two ERMS cell lines and demonstrated their resistance to cyclophosphamide, vincristine, and etoposide [41]. The study did not analyze the level of ALDH1 cell expression in tissue sections.

Aldehyde dehydrogenase (ALDH) is a superfamily of enzymes that participates in the key physiological processes that provide cell homeostasis and protect cells against the cytotoxic, mutagenic, and carcinogenic effects of aldehydes [42,43]. ALDH activity is crucial for retinoic acid formation, a factor that is essential for proliferation and differentiation processes [36]. The isoform ALDH1A1 serves as a marker for the identification and isolation of NSC and CSC [43]. The assessment of ALDH1 activity has been considered to be a reliable marker of CSC in malignant neoplasms of the head and neck [44], lung [45], pancreas [46], cervix [47], breast [48], prostate [49], bladder [50] and large intestine [51]. It has been proven that high ALDH1 expression is associated with the increased oncogenic potential of neoplastic cells with stem cell properties [43]. ALDH plays also a key role in CSC resistance to some anticancer drugs, such as cyclophosphamide [52], anthracyclines, taxanes [53], and bortezomib [54]. Moreover, high ALDH activity has been associated with resistance to radiotherapy [55]. In the presented study, the expression of ALDH1A1 was observed in all control group cases, i.e., in the healthy striated muscle tissue. The presence of ALDH has been demonstrated in many normal tissues of the human body, including skeletal muscle [56]. In response to trauma, toxins, or muscle degenerative diseases, striated muscles show the ability to regenerate, via muscle progenitor cells, which, upon activation, transform into actively proliferating myoblasts, which are cells that have been proven to have high ALDH activity and ALDH1A1 expression [57,58,59].

The quoted literature on the expression of CSC markers draws attention to the considerable diversity and discrepancy of the presented results on stem cells in RMS. Although immunohistochemistry is both a proven and widely used method both in clinical practice and in experimental research, the lack of standardization in the presentation of IHC reaction results and their interpretation makes it difficult to compare the published results of related studies [60]. The use of a semiquantitative scoring system allows for the translation of subjective and often descriptive results of the pathologist’s interpretation into quantifiable data, which are then subject to statistical analysis [60]. The semiquantitative IRS scale is a commonly used immunohistochemical analysis method for a broad spectrum of IHC markers [60]. There are few studies assessing the prognostic role of CSC markers in sarcomas. Walter et al. observed lower survival rates in ERMS patients with confirmed high CD133 expression (survival probability less than 50%). In most cases, CD133 expression was weak, moderate, or there was no color reaction in the tissue material. The survival probability for this subgroup of patients was approximately 75% (comparable to the survival rates of translocation-negative RMS patients). The authors of the study suggested the utility of CD133 as a prognostic marker in ERMS cases and as a potential therapeutical target in children [14]. Zambo et al. showed that there was a significant correlation between increased CD133 expression in RMS pediatric patients and shorter overall survival and event-free survival (relapse, progression, or death). CD133 expression was assessed on the basis of the percentage of positively stained cells, regardless of the staining intensity, and the cut-off point for high expression was 20% positively stained tumor cells [16]. Humphrey et al. confirmed that low CD44 expression (<40% positively stained tumor cells) correlated with a worse prognosis. Considering the fact that most of the ARMS cases presented with a low CD44 expression, it cannot be ruled out that the histopathological variant influenced the results of the survival analysis presented by the authors [24]. On the other hand, Heerema-McKenney et al. showed no statistically significant prognostic influence of CD44 expression on the overall survival or the time to relapse in RMS [26]. The potential role of ALDH as a biomarker and target for therapy in metastatic disease is currently under intensive research. The high expression of some ALDH isoforms in the cancer stem cells of various malignant tumors is correlated with a worse prognosis [36]. However, the prognostic significance of ALDH seems to be controversial. Discrepancies in the literature may result not only from the methodological differences, but also from the specificity of a given type of cancer and the differentiation degree of neoplastic cells.

## 5. Conclusions

The enthusiasm surrounding the stem cells theory stems from hope for the development of new and effective methods for targeted anti-cancer therapy. The surface and cytoplasmic markers determining specific SSC constitute a potential target for novel therapies. Although high expression of potential SSC markers such as CD24, CD44, CD133, and ALDH1A1 in RMS in children was confirmed in the presented study, the expression of these markers was not correlated with overall survival in the analyzed cohort. Further prospective studies on larger groups of patients are necessary to determine the role of SSC in RMS carcinogenesis and to define novel SSC markers specific to RMS. The first step towards this is the design of molecular preclinical and xenograft studies.

## Figures and Tables

**Figure 1 diagnostics-12-01895-f001:**
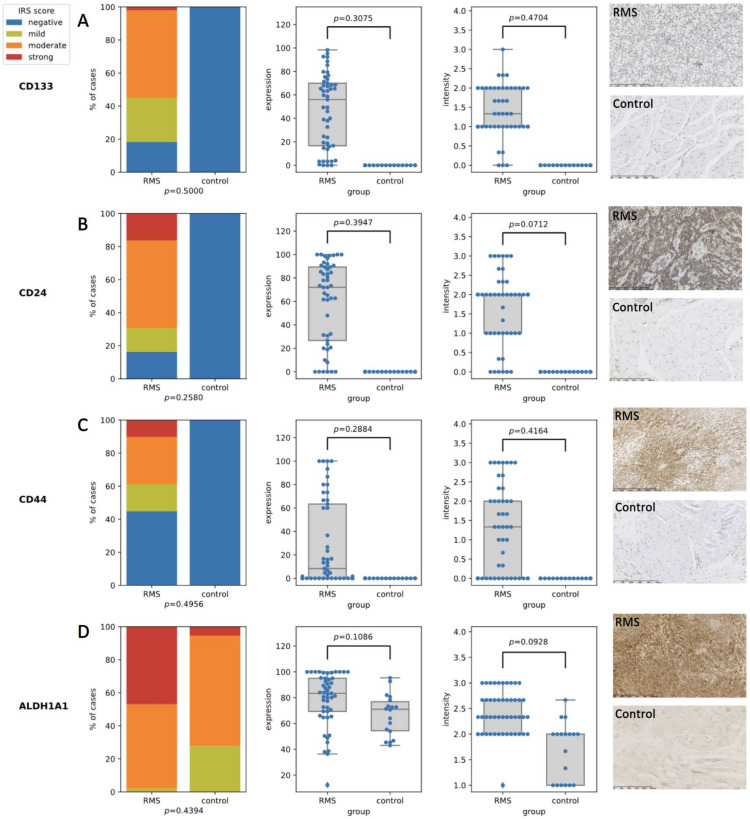
The expression of the stem cell markers in the RMS and normal muscles. (**A**) The expression of the CD133 middle panels—comparison between RMS cases and controls in terms of expression and intensity, right panel—the IHC staining of RMS case (upper panel) and control (lower panel) sample; (**B**) The expression of the CD24 middle panels—comparison between RMS cases and controls in terms of expression and intensity, right panel—the IHC staining of RMS case (upper panel) and control (lower panel) sample; (**C**) the expression of the CD44 middle panels—comparison between RMS cases and controls in terms of expression and intensity, right panel—the IHC staining of RMS case (upper panel) and control (lower panel) sample; (**D**) The expression of ALDH1A1 middle panels—comparison between RMS cases and controls in terms of expression and intensity, right panel—the IHC staining of RMS case (upper panel) and control (lower panel) sample.

**Figure 2 diagnostics-12-01895-f002:**
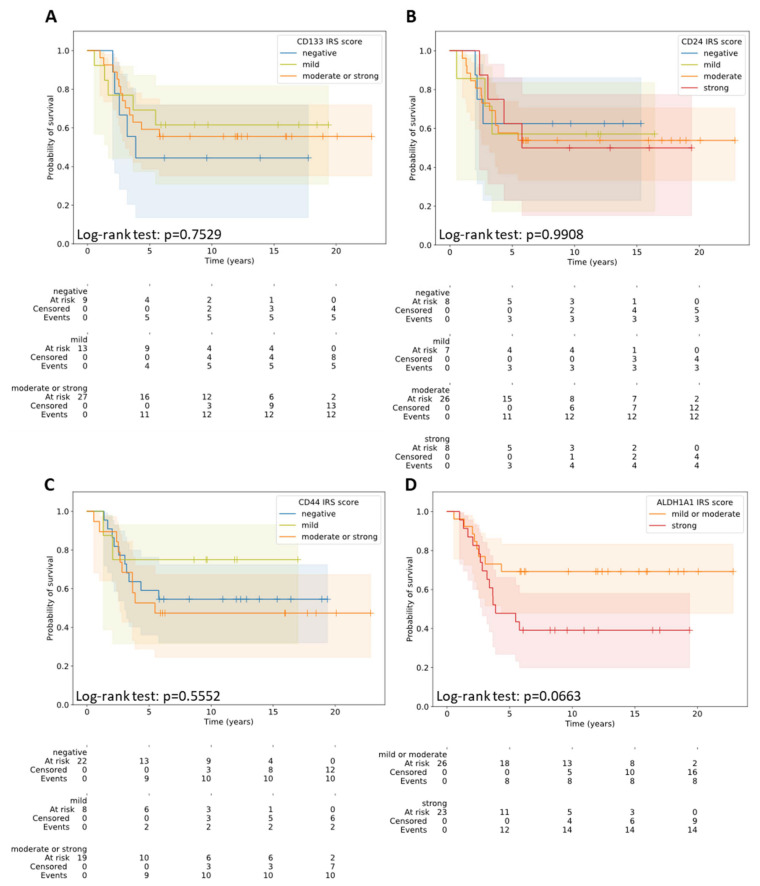
Overall survival in the patients with different IRS scores for CD133 (**A**), CD24 (**B**), CD44 (**C**), and ALDH1A1 (**D**).

**Table 1 diagnostics-12-01895-t001:** Baseline patients’ characteristics.

Variable		Cancer (*n* = 49)	Control (*n* = 18)	*p*
**Gender**	female	19	12	0.0554 ^a^
	male	30	6	
**Age (years)**	median (IQR)	4.8 (2.3, 8.7)	14.0 (8.5, 15.5)	0.0001 ^b^
**Histologic subtype**	ARMS	19 (39%)	-	-
	ERMS	30 (61%)	-	-
**Tumor localization**	favorable	17 (35%)	-	-
	unfavorable	32 (65%)	-	-
**Tumor size**	≤5 cm	16 (33%)	-	-
	>5 cm	33 (67%)	-	-
**T**	T1	15 (31%)	-	-
	T2	34 (69%)	-	-
**N**	N0	39 (80%)	-	-
	N1	10 (20%)	-	-
**M**	M0	32 (65%)	-	-
	M1	17 (35%)	-	-
**TNM stage**	1	15 (31%)	-	-
	2	2 (4%)	-	-
	3	15 (31%)	-	-
	4	17 (34%)	-	-

^a^ Fisher’s exact test; ^b^ Mann–Whitney U test.

**Table 2 diagnostics-12-01895-t002:** Expression of the CSC markers in T1 and T2 tumors.

			T1	T2	*p*
**CD133**	**Expression**	median (IQR)	58.7 (17.7, 68.3)	52.7 (17.2, 71.0)	0.4611 ^a^
	**Intensity**	median (IQR)	1.0 (1.0, 1.8)	1.3 (1.0, 2.0)	0.2062 ^a^
	**IRS score**	negative	3	6	1.000 ^b^
		mild	4	9	
		moderate	8	18	
		strong	0	1	
**CD24**	**Expression**	median (IQR)	83.3 (64.0, 90.0)	64.8 (21.4, 88.2)	0.0822 ^a^
	**Intensity**	median (IQR)	2.0 (1.5, 2.0)	2.0 (1.0, 2.2)	0.3596 ^a^
	**IRS score**	negative	1	7	0.3224 ^b^
		mild	1	6	
		moderate	11	15	
		strong	2	6	
**CD44**	**Expression**	median (IQR)	1.7 (0.0, 76.7)	10.8 (0.0, 60.0)	0.4119 ^a^
	**Intensity**	median (IQR)	0.7 (0.0, 2.2)	1.3 (0.0, 2.0)	0.3077 ^a^
	**IRS score**	negative	8	14	0.6649 ^b^
		mild	1	7	
		moderate	4	10	
		strong	2	3	
**ALDH1A1**	**Expression**	median (IQR)	81.3 (71.0, 92.0)	83.7 (66.8, 95.2)	0.4524 ^a^
	**Intensity**	median (IQR)	2.3 (2.0, 2.7)	2.3 (2.1, 2.7)	0.2544 ^a^
	**IRS score**	mild	1	0	0.3148 ^b^
		moderate	8	17	
		strong	6	17	

^a^ Fisher’s exact test; ^b^ Mann–Whitney test.

**Table 3 diagnostics-12-01895-t003:** Expression of the CSC markers in TNM stage 1 + 2 + 3 and TNM stage 4.

			TNM Stage 1 + 2 + 3	TNM Stage 4	*p*
**CD133**	**expression**	median (IQR)	57.3 (16.5, 68.4)	49.3 (23.7, 71.3)	0.722 ^a^
	**intensity**	median (IQR)	1.5 (1.0, 2.0)	1.3 (1.0, 1.7)	0.565 ^a^
	**IRS score**	negative	6	3	reference
		mild	8	5	0.515 ^b^
		moderate or strong	18	9	0.757 ^b^
**CD24**	**expression**	median (IQR)	62.7 (22.9, 87.8)	78.0 (67.0, 93.3)	0.363 ^a^
	**intensity**	median (IQR)	2.0 (1.0, 2.0)	2.0 (2.0, 2.3)	0.632 ^a^
	**IRS score**	negative	5	3	reference
		mild	6	1	0.168 ^b^
		moderate	17	9	0.578 ^b^
		strong	4	4	0.939 ^b^
**CD44**	**expression**	median (IQR)	12.0 (0.0, 73.3)	4.7 (0.0, 60.0)	0.634 ^a^
	**intensity**	median (IQR)	1.2 (0.0, 2.1)	1.3 (0.0, 2.0)	0.953 ^a^
	**IRS score**	negative	15	7	reference
		mild	4	4	0.191 ^b^
		moderate or strong	13	6	0.906 ^b^
**ALDH1A1**	**expression**	median (IQR)	83.0 (68.3, 99.1)	85.3 (72.3, 94.0)	0.715 ^a^
	**intensity**	median (IQR)	2.3 (2.0, 2.7)	2.3 (2.3, 2.7)	0.988 ^a^
	**IRS score**	mild or moderate	18	8	reference
		strong	14	9	0.697 ^b^

^a^*p*-value for TNM stage derived from separate GLM model for each variable in the table; ^b^*p*-value for TNM stage derived from separate multinomial logistic regression model for each variable in the table. Age and histologic subtype were included as covariates in all models.

**Table 4 diagnostics-12-01895-t004:** Overall survival prognostic factors.

Variable	Group	Median Survival (Years)	*p* (Log-Rank Test)
Gender	female	NA	0.7533
	male	NA	
Histologic subtype	ARMS	3.4	0.0279
	ERMS	NA	
Tumor localization	favorable	NA	0.0977
	unfavorable	5.5	
Tumor size	≤5 cm	NA	0.0134
	>5 cm	4.3	
T	T1	NA	0.0241
	T2	4.3	
N	N0	NA	0.0168
	N1	2.7	
M	M0	NA	0.0006
	M1	3.2	
TNM stage	1	NA	0.0045
	2	NA	
	3	NA	
	4	3.2	
CD133 IRS score	negative	3.9	0.7529
	mild	NA	
	moderate or strong	NA	
CD24 IRS score	negative	NA	0.9908
	mild	NA	
	moderate	NA	
	strong	NA	
CD44 IRS score	negative	NA	0.5552
	mild	NA	
	moderate or strong	5.5	
ALDH1A1 IRS score	mild or moderate	NA	0.0663
	strong	3.9	

**Table 5 diagnostics-12-01895-t005:** Prognostic value of the CSC markers in RMS.

Variable	HR (95% CI)	*p* (Cox Regression)
Age	1.11 (1.03, 1.21)	0.0095
CD133 expression *	0.94 (0.82, 1.07)	0.3256
CD133 intensity	0.71 (0.38, 1.35)	0.3026
CD24 expression *	1.03 (0.91, 1.16)	0.6487
CD24 intensity	1.31 (0.80, 2.14)	0.2771
CD44 expression *	1.03 (0.92, 1.15)	0.6391
CD44 intensity	1.07 (0.73, 1.56)	0.7257
ALDH1A1 expression *	1.06 (0.85, 1.31)	0.6272
ALDH1A1 intensity	2.44 (0.84, 7.13)	0.1018

* HR for increase by 10.

## Data Availability

Data are available upon request to the corresponding author.

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
