# Peer review of "Cancer Stem Cell Markers in Rhabdomyosarcoma in Children"

_diagnostics, 2022, doi:10.3390/diagnostics12081895_

Round 1

Reviewer 1 Report

The english language could be improved.

Author Response

Dear Reviewer. Thank you kindly for your work on our manuscript. 

Q: The english language could be improved.

A: The English language has been improved. The manuscript was corrected by the MDPI Language Editing Service.

Reviewer 2 Report

Review common of Radzikowska et al.

Rabdomyosarcomas (RMS) are a common cancer of children. In this study, the authors aim to investigate the correlation between cancer stem cell markers and RMS. Although there is no significant correlation between cancer stem cell markers and RMS, the authors still provide value information in the RMS field.

Minor points

1.     In Figure 1, to make the panel more intuitively to the readers, it is suggested to label RMS and control group nearby the IHC panels.

2.     The discussion of cancer stem cell markers could be more concise.

Author Response

Dear Reviewer. Thank you kindly for your work on our manuscript. 

1.English language and style are fine/minor spell check required

The English language has been improved. The manuscript was corrected by the MDPI Language Editing Service.

2.Not all of the cited references are relevant to the research

The references not relevant to the research have been removed.

3.In Figure 1, to make the panel more intuitive to the readers, it is suggested to label RMS and control group nearby the IHC panels.

Figure 1 has been corrected according to the reviewer's suggestions.

4.The discussion of cancer stem cell markers could be more concise.

The discussion of cancer stem cell markers has been shortened.

Reviewer 3 Report

A study to assess cancer stem cell markers (CSC) including CD24, CD44, CD133 and ALDH1A1 in rhabdomyosarcoma (RMS) in children and to define their prognostic roles. Several points should be noted as below.

1) As to IHC staining was assessed by two independent researchers.”, Of note, IHC staining needs to be assessed by two experienced pathologists independently other than (better than) researchers. Therefore, the expression levels of CSC markers needs to be re-analyzed to see whether these present results are changed. In addition, the dilution and the expression localization of CSC markers should be added.  

2) The study found that no prognostic relevance of the expression of either of the assessed CSC marker in patients. On the other hand, how about the prognostic and survival impacts about combination of these CSC markers in RMS patients? It is interesting to show something.

3) The 2nd paragraph in the Introduction section should be shortened.

Author Response

Dear Reviewer. Thank you kindly for your work on our manuscript. 

1. English language and style are fine/minor spell check required

The English language has been improved. The manuscript was corrected by the MDPI Language Editing Service.

2. The 2nd paragraph in the Introduction section should be shortened.

The 2nd paragraph in the Introduction section has been shortened.

3. The study found that no prognostic relevance of the expression of either of the assessed CSC markers in patients. On the other hand, how about the prognostic and survival impacts of the combination of these CSC markers in RMS patients? It is interesting to show something.

In the present study multivariate Cox proportional hazard model was performed. Variables with p<0.1 in univariate analysis were qualified to variable selection. The initial model contained only M trait (variable with the lowest p in the univariate analysis). At each step of the procedure, every remaining variable (not yet in the model) was examined and the one with the lowest p-value was introduced on the condition of p<0.1; variables were removed from the model when p-values became higher than 0.1. The variable selection procedure was ended when no further changes in the variable list occurred.

Moreover, two additional variants of the multivariate Cox proportional hazard model were performed. All selected variables gender, histologic subtype, tumor localization and size, T, N, M, TNM stage, TNM stage (4 vs. other), CD133 IRS scores, CD24 IRS scores, CD44 IRS scores, and ALDH1A1 IRS scores as well as the expression and staining intensity of the selected CSC markers, were included regardless of p in the univariate analysis. The result was the same as above. The prognosis was significantly worse in M1 than in M0.

When only CSC markers variables were selected, regardless of p in univariate analysis, the result showed a less favorable prognosis when ALDH1A1 expression was strong.  ALDH1A1 IRS score was included in the model with p=0.0735 and  HR was 2.22 (95% CI: 0.92-5.29). The result was therefore not statistically significant and the model with M is more suitable.

In conclusion, attempts to include more CSC markers do not generate a better model of survival, and still, the best we have in this data is the model with only  M trait. It may result from the fact that the disease is rare and the study group was too small.

4. As to “ IHC staining was assessed by two independent researchers.”, Of note, IHC staining needs to be assessed by two experienced pathologists independently other than (better than) researchers. Therefore, the expression levels of CSC markers need to be re-analyzed to see whether these present results are changed. In addition, the dilution and the expression localization of CSC markers should be added. 

I would like to inform that IHC staining in our study was assessed by two independent and experienced pathologists of the Department of Pathology of the Institute of Mother and Child in Warsaw, which is a leading Polish center for diagnostics of childhood cancers, in particular soft tissue and bone sarcomas. Dr. Teresa Klepacka, the researcher evaluating IHC staining in our study, is the head of the aforementioned Department and the pathomorphology consultant for solid tumors diagnostics on behalf of the Polish Pediatric Solid Tumors' Group. It would be difficult to find a better expert in Poland dealing with the diagnostics of  RMS in children.

The dilution and the expression localization of CSC markers have been added.

Round 2

Reviewer 3 Report

The authors had made the revision according to my concerns. No other questions.